# "Stay committed on the frontlines": sustainability of the activism of social workers in Guiyang, China

Junling Jin[1], Sai Tang[2], Yifu Liu[3]*, Xiaoyu Luo[1]

**1** School of Sociology, Guizhou Minzu University, Guiyang City, Guizhou Province, China, **2** School of Environment, Resources and Development, Asian Institute of Technology, Klong Luang, Pathumthani, Thailand, **3** College of History and Culture, Hubei Normal University, Huangshi City, Hubei Province, China.

* bestyifu@outlook.com

## Abstract

Social work has played an increasingly important role in social governance, service provision and driving for social change in China. Despite of the advancement of social work at the macro level in the country, substantial challenges remain to the sustainability of activism of social workers under the current social and economic context in China. Yet the fundamental reasons that underpin social workers' persistence in activism haven't been well investigated. By following the sustained commitment theory, the researchers investigated how and why frontline social workers sustain their activism over time. Qualitative research was conducted with 15 frontline social workers from local organizations in Guiyang, Guizhou Province, China. The findings highlight the role of "creativity" in maintaining activism of social workers in a way that both challenge and advance the sustained commitment theory. Instead of being confined to a fixed set of creative actions described by the theory, this study stresses the creativity of social workers in addressing challenges and generating feasible strategies in specific contexts. An essential prerequisite "confidence" for creativity in sustaining activism in social work was introduced. This study contributes to a deeper understanding of sustained activism in social work through the lens of social workers and the enhancement of the professional support for social workers in China.

## Introduction

Following the implementation of reform and opening-up policies, China witnessed the gradual dissolution of its traditional work-unit-based welfare system alongside the burgeoning market economy [1,2]. Social issues, such as the widening wealth disparities, an aging population, and rural poverty, etc., are becoming increasingly prominent [1,2]. In response, the concept of social work was reintroduced [3], with a primary focus on mitigating the social problems arising from the social

**Data availability statement:** Due to ethical considerations and participants' explicit concerns regarding potential re-identification risks, the original research data cannot be made publicly available. Despite implementing standard anonymization protocols, interviewees expressed apprehension that their identities might be discernible through narrative details in the dataset, leading to their formal objection to public data sharing. In compliance with institutional review board requirements and research ethics standards, access is restricted to de-identified data subsets. Researchers or institutions requiring data access for legitimate academic purposes may send requests to the corresponding author Dr. Liu (bestyifu@outlook.com). Alternatively, if Dr. Liu is unavailable, inquiries may be directed to Prof. Hu (54375559@qq.com, +86 151 8080 0912), a designated member of the research ethics review committee, school of sociology, Guizhou Minzu University. Please do not hesitate to contact us.

**Funding:** The author(s) received no specific funding for this work.;

**Competing interests:** The authors have declared that no competing interests exist.

transformations. In 1988, a significant milestone was achieved when the Hong Kong Social Workers Association collaborated with China mainland entities to establish the first social worker training centre in Shenzhen [4,5]. The 1990s saw the gradual restoration and development of social work education in universities, marking the beginning of the professionalization of social work in China [6].

The new millennium heralded a pivotal era for social work in China. In 2006, the Central Committee of the Communist Party of China issued the "Opinions on Strengthening the Construction of Social Work Professionals", followed by the establishment of the "Social Worker Professional Level Examination" in 2008 [7,8]. These developments led to an exponential increase in the number of professional social workers across the country. According to the "Statistical Bulletin on the Development of Civil Affairs of 2022", there were 1.16 million certified social workers by the end of 2022 [9]. The true number of social work practitioners is probably greater than reported and continues to rise steadily, driven by the profession's wide-ranging and multifaceted demands.

Social workers in China have played an increasingly important role in the realms of social governance and service provision [10]. Social work has served as a significant instrument for driving social change, enjoying a degree of legitimacy and operational convenience due to the governmental recognition as a professional field in the country., It is often integrated into the social governance framework through supportive policies and government procurement of services [11,12]. In the pursuit of social justice, many social workers are not only providing services, but rather practitioners of activism. Through advocating and providing services, they actively participate in and propel social change, for example, giving voice to marginalized groups [13,14]. This represents their activism which is another facet of social work beyond clinical practice, and serves as a potential force for societal transformation. It is evident that social workers are directly linked to the stable functioning and progressive advancement of society [15]. However, how social workers in China can persist in their activism under multiple pressure such as the resource constraint, cultural traditions and societal expectations, etc., remains a less investigated area.

While the development of social work in China has achieved significant progress at the macro level, it continues to face substantial challenges. Scholars have pointed out that the number of professional social workers remains insufficient and does not fully meet societal demands [16]. Additionally, the societal acknowledgment level of the social work profession and professional remuneration I are still inadequate [17,18]. Furthermore, a severe attrition of social work talents was also observed, which poses significant challenges to the retention and sustainability of the social work [19]. For example, government figures reveal that fewer than 40% of certified social workers continue practicing in the profession, underscoring systemic retention challenges in professional talents [9]. Surveys also show that social workers' turnover intension rate reaches as high as 50%, with many regions reporting turnover rates exceeding 20% [20,21]. The leading researcher of the present study has similar observations that many frontline social workers have left their positions in recent

years due to policy changes and decrease of projects, while serving as supervisor in local social work organizations and working in academia in the research area. For instance, in two social work organizations in Guiyang, there were a total of 55 staff members in 2021 and 2022; however, following the adjustment of administrations for the social work sector at both the national level (the establishment of the Society Work Department of the Communist Party of China Central Committee) and local levels that lead to program adjustment and reduction, there were only 7 full-time social workers remained by 2024.

The development and challenges of the activism of social workers under the current social and economic context in China raises the questions: Are economic factors the primary reason for social workers' persistence in activism? What are the fundamental factors that underpin social workers' commitment? To explore these questions, the researchers focus on 15 social workers from various fields in Guiyang who have steadfastly remained on the frontlines through semi-structured in-depth interviews. The researchers investigated how frontline social workers sustain their activism and critical factors in their perseverance under the guidance of the sustained commitment theory.

This study reveals the pathways through which social workers persist in activism and the complex dynamics driving their commitment. It contributes to the understanding of social workers' choices and agency from social worker perspective, offering detailed and practical insights into the development of social work, moving beyond the grand narratives of the general development of the sector. In the literature review section, the researchers first draw on relevant literature and key concepts to clarify the strong connection between social workers and activism. Then, researchers conducted a systematic review of academic attention and development on collective actions before introducing the sustained commitment theory. The methodology section introduces the research area, subject selection, data collection, and analytical methods. The analysis section presents the main findings, providing a thorough analysis of the data. The discussion section summarizes the research results, discusses the theoretical and practical significance of the research, and offers recommendations while acknowledging the study's limitations.

## Literature review

Social workers are professionals who utilize specialized knowledge, skills, and ethical principles to serve individuals, groups, and communities, aiming to promote their well-being [22]. They are dedicated to helping people address various life challenges, improving social functioning, and advocating for social equity and justice [23,24]. With a professional mission to serve society and enhance the lives of others, social workers are a vital force in driving social progress and achieving a fairer society [25].

The philosophy of social workers aligns extensively and profoundly with activism. Activism refers to the proactive actions or organized efforts undertaken by individuals or groups to drive change in social, political, economic, or environmental domains [26]. It can manifest itself as support for issues such as social justice, equal rights, and environmental protection, or as opposition to specific injustices or inequities [27]. Regardless of its various forms, the goal of activism is to spark societal discussion, propel actual change, and achieve specific social objectives or values [28]. It is evident that social work and activism share a common goal, particularly in advocating social justice, promoting policy reform, and supporting vulnerable populations.

Scholars in the last century attempted to formulate a model for activism in social work [29]. The fundamental values and principles underlying social work practice are seen as highly overlapping with activism and are often adopted in advocates for social justice [30]. Subsequent scholars agree to such opinion and call for an emphasis on the role of activism in social work [31]. Particularly, establishing the foundation of activism in social work was advocated to better address complex social issues meanwhile ensuring the credibility and influence of social workers [30]. Then, the sustainability of the activism of social workers becomes an issue that cannot be ignored.

However, there is a lack of understanding of sustainability of activism. The absence of understanding on sustainability can be traced back to the early studies on collective action that primarily focused on irrational behaviours of individuals,

especially during wartime. The resource mobilization theory that emerged in the 1960s systematically analysed why people participate in collective actions aiming at driving social progress, yet its analytical lens predominantly centred on the acquisition and management of resources by social movements at the macro level [32]. This theory also introduced a paradox known as the free rider problem, where individuals seek to benefit from collective actions without directly participating [32,33]. Subsequent new social movement theory shifted their focus from structural factors to culture and identification [34]. Scholars argue that collective action itself is a process of subculture construction and identity formation [34,35]. While new social movement theory foregrounds participant agency and subjectivity, it nevertheless falls short of elucidating the mechanisms driving the formation and persistence of collective action, leaving critical gaps in understanding how such engagement crystallizes and endures.

Expanding the conceptual terrain mapped by earlier scholars, Downton and Wehr (1998) formulated the sustained commitment theory to interrogate the mechanisms underlying long-term activist engagement. They explained that commitment to activism is inextricably linked to sustained action in the face of adversity [36]. Detailed and extensive factors have been examined and identified, including integrating activism into daily life, holding beliefs that support proactive activism, feeling closely connected to the organization, fostering opportunities for action, sharing a vision of peace with like-minded individuals, managing responsibilities, criticism, and burnout, etc [36,37]. The theory highlighted and concluded that creativity is the underlying nature of these factors and thus the most critical in sustaining activism [36]. Persistent activists are rational in choosing their course of action and demonstrate creativity in shaping their lives, managing their commitments, avoiding burnout, designing and implementing projects [36,37]. This study will follow the model provided by the sustained commitment theory, to test whether a variety of creative behaviours promote the sustainability of activists among frontline social workers in Guizhou, China, and further attempt to uncover the fundamental reasons that enable them to overcome difficulties and persist.

Considering the distinct cultural, institutional, and social norms in Guiyang within the Chinese context, this study also introduces the cultural opportunity model in analysis [38]. Due to the differences in cultural and ideological factors, the forms of collective action often vary significantly [39]. The success of activism is also closely tied to shifts in the cultural environment and the emergence of cultural opportunities [40]. Activists must adeptly identify environmental changes and devise strategies to leverage cultural opportunities, thereby advancing their movements [41]. This approach helps observe and understand the unique methods and strategies of individual activist [42], enabling a shift from studying standardized collective action practices to analysing the particularities of local activism.

## Methodology

Guizhou is a key province in the country's Great Western Development Strategy located in the southwestern part of China [43,44]. With its city GDP ranking sixth in western China and sixtieth nationally, its socioeconomic development trajectory epitomizes the characteristic of China's central and western provinces [45]. Following the footsteps of China's coastal regions, the development of social work in Guizhou Province has shown a positive trend [46]. The study is conducted in Guiyang City, the capital of Guizhou Province with population of over six million [47]. According to reliable data in 2022, there were 67 social work service organizations, 1,811 certified social workers, and 6,182 social work professionals in Guiyang [48]. The researchers conducted qualitative research on full-time frontline social workers in social work organizations in Guiyang City.

In 2024, against the backdrop of program and funding reductions, the researchers interviewed social workers who are still committed to activism. After obtaining the approval of the ethics review, the researchers officially launched the fieldwork on March 16, conducted the first face-to-face interview on April 28, and completed all data collection by September 1, 2024. To ensure the representativeness and diversity of the respondents, the researchers made efforts to establish research relationships with frontline social workers from various fields and different organisations. These individuals served on the frontlines for at least one and a half years, despite income decrease. The researcher visited 11 social work

organizations, communicated with potential participants, selected those who met the selection criteria, and obtained their consent for participating in the study. Ultimately, research relationships were established with 15 respondents from 9 organizations. Their years of experience, age, and fields of activism are listed in Table 1 below. The researchers had planned to conduct a 1.5-hour in-depth interview with each respondent. In practice, the interview duration was slightly extended to an average of 1.78 hours per respondent, with some respondents requiring an additional session of interview.

The study employed semi-structured interviews; the researchers completed all interviews over a four-month period. Interviews were conducted in secure third-party venues and recorded using a non-internet-connected audio recorder and encrypted for storage. The recordings were transcribed into text by the researchers. Thematic analysis was applied for its advantage of flexibility yet high structuredness, to present the data and results clearly [49]. Given the absence of directly

**Table 1. The information of respondents from field.**

| Respondent NO. | Age | Years of Experience | Work field | Work contents |
|---|---|---|---|---|
| 01 | 42 | 12 | Environmental Social Work | Forest and Ancient Tree Conservation, Environmental Policy Advocacy, and Volunteer Team Building |
| 02 | 27 | 4 | Geriatric Social Work | Case Support and Case Management, Policy and Legal Awareness Promotion, Crisis Intervention |
| 03 | 36 | 10 | Petition Social Work & Geriatric Social Work | Case Support and Case Management, Social Governance, Data Analysis and Policy Recommendations, Chronic Disease Management for the Elderly, and Emergency Rescue for Missing Elderly Individuals |
| 04 | 30 | 6.5 | Community Correction Social Work | Case Management, Case Counselling, Vocational Training, Employment Assistance Projects, and Crisis Intervention |
| 05 | 38 | 14 | Child Social Work | Left-Behind Children Programs, Case Counselling, Case Management, Crisis Intervention, Family Relationship Intervention, Community Activity Organization, and Academic Tutoring projects |
| 06 | 28 | 4 | Geriatric Social Work | Case Management, Social Adaptation and Life Skills Training, Chronic Disease Intervention, Family Linkage Projects, and Elderly Volunteer Development |
| 07 | 24 | 2 | Anti-Drug Social Work | Drug Prevention and Awareness Education, Addict Rehabilitation, and Reintegration Support into Society |
| 08 | 25 | 2.5 | Community Correction Social Work | Case Management, Crisis Intervention Projects, Community Building, and Activity Organization |
| 09 | 33 | 9 | Youth Social Work and Left-Behind Children Programs | Academic Encouragement Initiatives, Educational Resource Linkage, Fundraising, Public Relations (especially Government Relations), Policy Recommendations, Rural Livelihoods, and Women's Empowerment |
| 10 | 40 | 15 | Youth Social Work and Left-Behind Children Programs | Educational Support Projects, Guardianship and Legal Protection Projects, Policy Advocacy and Policy Recommendations |
| 11 | 26 | 2.5 | Environmental Social Work | Environmental Monitoring, Environmental Policy Advocacy, Public Education, and Environmental Awareness Enhancement |
| 12 | 32 | 5 | Family Social Work | Women's Empowerment, Domestic Violence Intervention, and Legal Aid |
| 13 | 28 | 5 | Medical Social Work | Case Management, Social Resource Linkage and Support, Promotion of Medical Social Work, Policy Report Writing, and Full-Time Mother Parenting Support |
| 14 | 42 | 14 | Anti-Drug Social Work | Legal Aid, Drug Prevention and Awareness Education, Public Advocacy, and Policy Advocacy |
| 15 | 33 | 6 | Youth Social Work and Migrant Youth Programs | Children's Living Environment Improvement, Fundraising, Policy Advocacy, and Educational Support |
| Average | 32.3 | About 7.5 | | |

related or similar prior research, the researchers applied open coding, directly extracting codes from the textual material based on the language used by the respondents.

The leading researcher of the present study is a local university lecturer whose research motivation stems from her observation of the issue. Her first advantage in doing the study is her understanding of the activists and social workers in this field. Another significant advantage is her native proficiency in the local dialect, which allows her to carry out the study without language barriers. Her position as an external supervisor at two local social work organizations provides the advantage of connecting to various social work organizations in the area more easily and establishing links with potential respondents. To mitigate the potential ethical risks that researcher's multiple identities might bring and ensure the objectivity of the research, the independent researcher status was prioritized. The study did not select social workers who have personal connections with the leading researcher as respondents; interviews with respondents who have been acquainted with the leading researcher were conducted by the fourth author. The researchers signed written informed consent forms with all the respondents before the interviews, and the entire research process strictly adhered to the confidentiality protocols. The majority (12 out of 15) of the respondents preferred to remain anonymous, so all respondents were anonymized in the study, their identifiers have been replaced by numerical codes by the researchers.

## Analysis

The objective of this study is to explore the factors that contribute to the persistence of social workers' activism, specifically, how social workers sustain their activism over time. This exploration will enable the academic community to gain a deeper understanding of social change through the lens of social workers.

### "Creativity" for sustained actions being challenged?

In this study, many factors conventionally believed to be crucial for the persistence of activists appear to be ineffective. Numerous concepts and factors highlighted by the sustained commitment theory as promoting sustained action were either overlooked or denied by the respondents, such as: 1) creatively responding to competition; 2) creatively mobilizing and integrating resources; and 3) creatively managing personal life.

During the interviews, social workers extensively discussed the difficulties and challenges they encountered in their actions, but the keyword "competition" was rarely mentioned voluntarily by the respondents. When the researcher inquired particularly, respondent 04 provided an intriguing and philosophical response: "Competition, it both exists and does not exist." He believed it exists because:

> There must be competition. As you know, resources are so scarce. As we mentioned earlier, many people can't continue because there are no (mainly referring to government-purchased) projects, and projects means resources... But I always feel that the "competition" in this sector is not as blatant as in other sectors. Perhaps this is determined by the nature of this sector... Maybe there is competition at the organization level, but it's not that intense.

He thinks it doesn't exist because:

> The main reason is that social work is still a "blue ocean" with many possibilities to "explore and expand". Having worked in this field for so many years, I've never felt the kind of competition where someone else's success gives you a sense of crisis, pushing you to do better, like in the corporate world. Although it's said that social work has developed for nearly 20 years (in China), it's still in its infancy. There are so many (new) things that we (social workers) can try, especially those with Chinese characteristics. For example, the works done in the ethnic areas in Guizhou, or in the rural and mountainous regions, these are Chinese characteristics social works that cannot be found in the West.

Overall, they believe that competition is inevitable and ubiquitous, but it will not lead to the elimination of individuals in the sector, hence it is not fundamental. Participant 04 used the business concept of the "Blue Ocean Market" [50] to metaphorically describe the local social work sector as an emerging market with minimal competition and great potential. Unlike the "Red Ocean Market" characterized by defeating competitors through competition, the "Blue Ocean Market" requires more innovation and exploration [50]. Therefore, the respondents feel that at this stage, competition is almost non-existent for them, as "there hasn't been an opportunity to demonstrate the level of competition in this area" (respondent 13).

As for innovation and exploration, the respondents candidly expressed their "lack of creativity in this aspect" (respondent 10), especially in terms of resource integration. As respondent 01 stated:

> ...we haven't been able to come up with better fundraising strategy. After all, funds from government-funded programs are the key. There are funds from other programs, but these funds are "like mosquito legs" (which means far from sufficient) …

The respondents held a rather negative view of their ability to mobilize resources. They emphasized that "fundraising and identifying potential donors for programs" which is characteristically referred as "finding programs", are the essential basis for the survival of organizations and social workers. In local language, earning money and fundraising are referred to as "finding money", which is equivalent to "finding program". The inability to "find money or program" is often attributed to social workers themselves, as respondent 11 expressed:

> It is still because of our inability. As you see, what we're doing is quite meaningful. If you ask around or conduct surveys, everyone supports it, including corporates, and common people. But when it comes to getting funds from them, it does not work. You may say people are selfish? Yes, that might be one reason. But surely, there are people genuinely willing to support. I think it's just because we haven't found the right people or the right approach.

Many of them have learned about "the fundraising models of others" (foreign social work organizations or NGOs) (respondent 01) through various means and have highly praised these models. They acknowledge the effectiveness of foreign fundraising models but find these models unsuitable for the local context. Several respondents believe that "there must be fundraising methods suitable for China or the local area" (respondent 11), but they have yet to discover them.

For personal aspect, the social workers who persist all consider themselves not to be good managers of their personal lives. They describe their lives as "not a complete mess, but certainly not good" (respondent 12). They lack both the emphasis and the actual investment in their personal lives. This is contrary to the point that activists can manage their personal lives well to avoid burnout in the sustained commitment theory. When discussing personal lives, the respondents exhibited a kind of troubled indifference:

> Our income as you see is low, and I don't expect it can get much better. I must thank Taobao (online shopping app in China), because it indeed helped me to lower the cost of living in the past few years.... But if you choose to be a social worker, you must accept the fact that you cannot make a lot of money… this is the sacrifice to make… (respondent 05)

> Being a social worker does not get paid well, but it requires a lot of investment (laughing jokily) …I will turn 35 in less than two years, but I am still unmarried, have no kids and no partner. I ask myself why I am living in this status... Working as a social worker is very different from working in a company and get paid stably. You must put your heart into it, otherwise you can't do it. (respondent 12)

From the accounts of the respondents, it is evident that they are not satisfied with their personal live status, yet they maintain a composed attitude that seems to take everything in stride. Economically, the standard of living continues to rise with economic development, and the income from social work is insufficient for frontline social workers to "live a better life" (respondent 05). On the other hand, doing this job well requires not only time but also a significant dedication of energy.

This leaves them so exhausted that they have little time to manage their personal lives, even major life aspects such as marriage and having kids. The social workers who persist refer to this as a voluntary "sacrifice" and accept it. They also say, "I don't know when I will quit," but even if they do quit, "it will be for other reasons" not because life has become untenable. As respondent 14 put it:

> Everyone wants to live better, orderly, stable, and don't need to worry about food and clothing. If you quit halfway because this job cannot offer that kind of better life to you, it can only mean that you weren't suited to be a social worker in the first place, not that you couldn't continue because the job cannot offer a better life to you. (respondent 14)

As the analysis presented above, the crucial elements that promote the persistence of social workers in the sustained commitment theory model, "creativity" or "ability" as described by the researchers, were not corroborated by the respondents in this study. Respondents either expressed their lack of creativity in mobilizing resources and managing personal life, or considering creativity is not so urgently required currently for the "competition" that is yet to become fundamental. This leads the researchers to doubt the relevance of this theory in the context of the study and whether creativity truly plays a pivotal role in the sustainability of activism more than other factors. This doubt was dispelled in the discussion of participants' creativity presented in real frontline contexts from different aspects.

### "Creativity" in real frontline contexts

As the interviews delved deeper, the creativity of the persistent social workers in their actions gradually unfolded in their narratives.

They not only achieved personal growth creatively, initiated and designed projects innovatively based on real needs and in specific context but also managed their relationships with the government in a creative manner. All of these contributed to their persistence on the frontline of actions for longer.

To put it straightforward, to achieve personal growth in work is a significant subjective reason why social workers are willing to stay on the frontlines. Its importance lies in that it allows individual social workers to experience their life better. The willingness to "keep learning and growing" which means to "create", will not only help them achieve personal growth but enjoy a better and fulfilling life. One respondent mentioned that many people at his age "seem to be alive, but in fact, are already dead" (respondent 05), because he thinks that many middle-aged people believe there is no new possibilities in their life, thus they cease to learn and grow again. But "being a social worker is different", as it allows one to "always feel that they are growing and need to keep learning and growing" (respondent 01). The respondents described their personal growth as beautiful experiences in different ways. Their descriptions of personal growth are the demonstration of "creativity" that the respondents both develop and deliver in their frontline works.

Moreover, achieving personal growth creatively is also described as an ability and competency in real frontline context. As being a social worker is "hardly ever repetitive work; there's always something new to learn and do" (respondent 09) and "If you can handle the work of a social worker, other jobs are not a problem anymore" (respondent 08). As respondent 04 described from another angle:

> I believe that it is an ability to grow in actions. I've seen many volunteers, students, and even full-time social workers in the sector, who are only finishing whatever they are assigned, but lack the ability to actively learn new things or skills in work... Naturally, they leave over time….

Respondent 04 views the attainment of personal growth as an ability, and not everyone can naturally acquire personal growth in certain environment. The ability to achieve personal growth creatively implies the competency of performing frontline work over the long term.

Furthermore, the respondents think it's not enough to be able to achieve personal growth; one must be able to create new initiatives or explore new possibility in specific contexts too. As respondent 04 put it, "the process of continuous doing is the process of continuous learning and exploring" (respondent 04). Personal growth can never be achieved if one neither learn nor "explore". Respondent 03 thinks that it is not right if "social workers in the country only know to 'draw the dipper by the gourd' in work (meaning "to imitate or follow others or other's work mechanically without independent thinking")'. Although the respondents blamed themselves for not having developed good approaches for fundraising by learning from the foreign models in previous discussion, this does not diminish their pride in their professional creativity. They believe that, in fact, exploring new initiatives and new strategies in local Chinese context expanded the space for their own and their organization's survival, as expressed by respondent 15:

> As you can see, social work in China mainly conducts conventional and old-fashioned (learned from foreign experience) works, such as case work, group work, school-based services, and medical social work, etc. So, I have always thought that social work is a foreign business that comes in China. But many of China's problems do not exist in foreign countries, there is no former foreign experience for these local problems. For example, the migrating population issue and let-behind children and elderly issue brought by the rural-urban labour migration in the country. These local social issues demand intervention from social work. And it requires proactive actions and exploration of social workers.

What respondent 15 refers to as "conventional and old-fashioned" pertains to the working strategy and content, specifically those working ways and modes that are taught in textbooks and directly learned from foreign models. They believe that only by aligning social work actions with real social issues and the needs of the people, can survival space be created for social workers and their organizations. Participant 15 illustrates this with his project example:

> Take the children's living environment renovation program I've been working on for example, it has already become a flagship initiative for local social work. I am confident that the government will continue to support it, and even if the government stops (funding), there will be other foundations willing to fund it.

The background of this program is that several years ago, there was a tragic incident where migrant youth lit a fire in a trash bin to keep warm during winter in this province, resulting in a collective poisoning. This incident caused a very negative social influence. The program that respondent 15 is doing directly addresses the survival and living conditions issue of migrant youth that local government and the society are concerned about.

Beyond practical work, the respondents also emphasized the necessity and their own capability to manage relationships with the government effectively. Half of the respondents listed policy advocacy or policy recommendations as part of their work and held a positive attitude towards maintaining good relations with the government.

> No matter how mystifying that social work has been described, at its core, it's about making people's lives better. Essentially, it aligns with the government's objectives, so I don't understand why some people always find some topic or initiatives that social work should cover and address to be politically "sensitive" issue.

Respondents do not perceive the relationship between the government and themselves and their organizations as a simple client-service provider dynamic, but rather as a "two-way guidance" relationship (respondent 04):

> To develop social work, government should be the lead, otherwise I wouldn't have had the opportunity to study social work at university; but when it comes to real practice and action, specifically "what to do" and "how to do", I always feel that we (frontline social workers) are the ones that are taking the lead. Though we (frontline social workers) are also "crossing the river by feeling the stones". Many issues are solved or advanced by us (frontline social workers)

through step-by-step exploration and practices in situations of uncertainty, and then we offer recommendations to the government.

Social workers believe that although they receive government funding, their relationship with government is not simple funder and recipient. But rather they are collaborating with the government to advance activism, to improve the welfare of the people and to promote social development. To some extent, they are the ones "who take money (from government) to discover problems" (respondent 13). In terms of working strategy, as concluded from the descriptions of the respondents, reliable data, representative and impressive stories, and a trustworthy organization with good reputation are the most effective support in collaborating with the government. Besides, taking proactive initiative to maintain a good relationship with the government for collaboration is another demonstration of social workers' creativity in real frontline context. For example, respondents 13 and 01 voluntarily write monthly reports and send them to relevant government departments.

As concluded from the analysis, it is evident that persistent activists do demonstrate considerable creativity from certain perspectives, as the sustained commitment theory suggested. The creativity in their personal growth, context-based exploration in new initiatives and government relationship management is the development in the sustained commitment theory. It provides more nuanced understanding for social workers' persistence of activism in China.

### "Confidence" as prerequisite for creativity in activism sustainability

Surprisingly, there is a breakthrough in explaining activism sustainability: social workers candidly state that their ability to persist in activism over the long term is due to their confidence; in other words, "those who cannot persist (in activism) lack self-confidence" (respondent 06). Confidence is considered "not necessarily more important than money, ability, or other aspects, but as a prerequisite for the persistence of social workers" (respondent 08). Without confidence, activism does not stand "from the starting point" (respondent 06). Thus, confidence as a prerequisite is introduced for the creativity in sustaining activism for social workers in this study. The respondents view the confidence they speak of from two angles.

Firstly, confidence in oneself. Those who persist in the long term see themselves as pioneers and individuals who are unique. They not only consider what others can offer them but value more the intangible gains that can be achieved through activism. They even believe in their ability to lead the sector and their organizations forward. The expression of respondent 09 is quite blunt on this point:

> For other ordinary work, it is nothing more than labouring. You do the work and get paid, which makes one feel like a tool... (laughing loudly). Right, there are also many who don't even work but still get paid; those people are like parasites, but the fact is that many people want to be this kind of "parasites". Either to be a tool, or a parasite... However, doing social work requires the courage to be "unordinary" or "unconventional" from what most people would do. It is like starting a business not for the sake of making money, and you must believe you can create something, or otherwise it's hard to keep doing it.

Secondly, confidence in the social work profession itself. When experienced social workers discuss the issue of talent attrition, they generally attribute this issue directly to practitioners' lack of sense of professional identity and confidence in the development of this profession. Despite the continuous introduction of policies by the state to promote the construction of the social work talent team, young people remain sceptical about the career prospects as a social worker. This reflects their uncertainty about the value and role of social work. For example, respondent 02 said:

> From the perspective of my college classmates who did not pursue or persist in social work, I can understand why they didn't choose to be a social worker or didn't persist in this profession. Because they are still not optimistic about the development of this profession. From enrolment to graduation, it's evident that most people do not identify with this

major, and many who initially took up social work jobs did so because they couldn't find other jobs. Usually, they chose to leave the profession once they had other options.

In summary, if creative action is the key to social workers' commitment to persistence, the data from this study supports the addition of "self-confidence" as a prerequisite for the key role of "creative action" in the persistence of activism.

## Discussion

This article focuses on the extension of social workers' careers and the sustainable contributions of their activism to social progress. The sustainability of social workers arises from their ability to leverage their creativity within their specific contexts, rather than being confined to a fixed set of creative actions. By adopting a variety of unique and innovative approaches, they make their activism practices possible. At the same time, on a conceptual level, social workers must possess a strong belief in their own creativity and trust in their potential in the profession. Only when this intellectual confidence is in place can their behavioural creativity truly manifest, enabling social workers to sustain their activism more effectively.

The study highlights creative actions that contribute to the persistence of activism, which aligns with the sustained commitment theory. At the same time, it reveals that certain factors emphasized by this theory are contradicted by the data from this study. In other words, the key to the sustainability of activism is not certain sets of creative behaviors. What truly matters is whether the creativity of social workers and activists is sufficient to enable them to address the challenges and difficulties within their environments effectively and further generate feasible and critical creative actions and strategies.

The findings of the study both challenge and advance sustained commitment theory. The model and key factors proposed by the sustained commitment theory is not universally applicable across different cultural and regional contexts. In Guiyang, the researchers observed and proposed a possibility, that is, the cultural opportunity structure defined by state-society synergy and collectivist pragmatism channels creativity toward adaptive strategies that ensure political legitimacy. This structural distinction may explain why the core assumptions of the sustained commitment theory require recontextualization based on specific contexts. This reflects activists' capacity to deploy their modes and orientations of action in response to shifting circumstances and contexts. It also reminds researchers to maintain heightened sensitivity toward the cultural opportunity structures of different regions in China. But the general logic of the sustained commitment theory applies to the finding of the study in the way that those who persist do exhibit remarkable traits in creativity. Building on this, this study introduces an essential prerequisite "confidence" for the positive correlation between creativity and sustainability in activism. The "confidence" pertains not only to their own abilities and creativity but also to the prospects of social work activism. Activists hold a high regard for their role. They don't think they are only "tools" in the workplace and the social systems but pioneers and changemakers of situations. This highlights the impressive agency that persistent social workers demonstrate. In other words, without confidence and personal will, creativity alone is insufficient to effectively promote activism sustainability.

In contrast to other related studies in China, this research uniquely provides an in-depth frontline perspective of social workers, differentiating itself from the predominant angles in Chinese scholarship. Despite China's significant regional diversity, existing studies predominantly focus on external conditions that affect activists' retention, such as wages, job availability, social status, work pressure, and career advancement [51,52,53]. These works advocate for improving these external conditions as solutions to the attrition problem of frontline social work talents [51,53,54]. While researchers posit that enhancing external factors can yield macro-level resolutions, this study introduces a distinct activist-centred lens, emphasizing how committed frontline social workers conceptualize career sustainability. This approach advances academic understanding of how subjectivity shapes committed activists, thereby contributing to broader social transformation.

Theoretically, this study not only explores the sustainability of activism in the Chinese context but also uncovers the key mechanisms that underly it. It addresses the theoretical gap in the intersection of social work and activism, particularly

practices and explorations in the Chinese context. The study not only introduces the sustained commitment theory but also identifies factors that matter for activism sustainability in local context. By adopting the first-person perspective of social workers, the internal motivations behind their choices and actions become more traceable. These contributions lay a solid foundation for future research.

The findings will also serve as a guiding basis for enhancing the sustainable development of the social work sector. This study provides critical insights into how frontline social workers in China can sustain their activism over the long term and drive social development. Beyond increasing budgets, enhancing wages and working conditions across the social work sector, in aspects of talent development and capacity building, greater emphasis should be on fostering the creativity of social workers. For example, in response to the problem of high certification rates but low retention rates among social workers, the talent selection process and social work qualification examinations should shift toward identifying individuals with creativity. Simultaneously, for current practitioners, particularly those that lack specialized backgrounds, governments and employer organizations could explore procuring professional courses from universities to provide free continuing education to them, to enhance both their professional competence and creativity. Moreover, building confidence is even more important than developing skills. Individual social workers' confidence in themselves and their confidence in the social work profession should be integrated. Government authorities should consider establishing systematic recognition mechanisms, supported by media advocacy, to bolster social workers' confidence and professional pride. Additionally, governments should formulate stable medium- to long-term development plans and make separate budgets for the sector and ensure the transparency of the development and budget plans. Through these, to reinforce practitioners' trust and confidence in the profession. In conclusion, it is essential to help social workers build multi-dimensional confidence from aspects of career development, personal growth, and the realization of their own value, etc. But, firstly, building confidence in the development of the social work profession should be the starting point.

In summary, this study not only offers new perspectives for the development of relevant theories but also contributes to enhancing the professional support system for social workers. It helps them better sustain their practices, thereby promoting the sustainable development of social work in China.

## Limitation

The study has inherent limitations, which are reflected in one research method, one research location, and one perspective. First, in terms of research methodology, the inherent limitation of qualitative research methods decides that it cannot capture the whole picture [55]. While subjective viewpoints are important, the extent to which common perspectives from a limited sample can be generalized and considered representative to wider context remains questionable. This is also decided by the nature of inherent limitation of the research method. Given the substantial size and growth potential of the social work workforce, the research team plans to integrate quantitative research in future research to both validate the key themes identified in interviews and enhance the credibility of the findings from this study. Second, this case study is based solely on frontline social workers in Guiyang. Clearly, the study results cannot represent the situations in economically more developed coastal regions in the east or underdeveloped rural areas. However, as the provincial capital of Guizhou province, Guiyang retains its representativeness among mid- to large-sized cities in China's central and western regions. Moreover, this study establishes a valuable reference for future comparative research. The transferable analytical framework could potentially be applied to other regions to conduct interregional comparisons, such as between urban and rural areas, as well as comparison with western cities or more developed coastal areas. This is not only the next step of the research team, but also a call for other researchers to adopt more diverse and broader sample selections to extend, validate, and challenge the conclusions of this study. Third, this study only adopts the perspective of social workers who have persisted in their careers. However, the viewpoints of those who did not persist are equally valuable. Although there must be challenges in sample selection and research relationships establishment, the researchers' call for research from the above-mentioned opposing perspectives could effectively complement and enrich the one-sided views presented in this study.

                                                                                                   

## Author contributions

**Conceptualization:** Junling Jin, Sai Tang, Yifu Liu.

**Data curation:** Junling Jin, Xiaoyu Luo.

**Formal analysis:** Junling Jin, Sai Tang, Yifu Liu.

**Investigation:** Junling Jin, Xiaoyu Luo.

**Methodology:** Junling Jin, Sai Tang, Yifu Liu.

**Project administration:** Junling Jin.

**Resources:** Junling Jin.

**Supervision:** Junling Jin, Yifu Liu.

**Validation:** Junling Jin, Xiaoyu Luo.

**Writing – original draft:** Junling Jin, Sai Tang, Yifu Liu.

**Writing – review & editing:** Sai Tang, Yifu Liu.

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
