## [Decision Letter · Decision Letter 0]

2 Mar 2025

PONE-D-25-03942“Stay Committed on the Frontlines”: Sustainability of the Activism of Social Workers in Guiyang, ChinaPLOS ONE

Dear Dr. Liu,

Thank you for submitting your manuscript to PLOS ONE. After careful consideration, we feel that it has merit but does not fully meet PLOS ONE’s publication criteria as it currently stands. Therefore, we invite you to submit a revised version of the manuscript that addresses the points raised during the review process.

Generally, the reviewers were concerned with generalizability of the findings reported in this version of the manuscript, as well as the language issue. Also, please consider other comments and suggestions for improvement. 

We look forward to receiving your revised manuscript.

Kind regards,

Tatchalerm Sudhipongpracha

Academic Editor

PLOS ONE

Journal Requirements:

3. In this instance it seems there may be acceptable restrictions in place that prevent the public sharing of your minimal data. However, in line with our goal of ensuring long-term data availability to all interested researchers, PLOS’ Data Policy states that authors cannot be the sole named individuals responsible for ensuring data access (http://journals.plos.org/plosone/s/data-availability#loc-acceptable-data-sharing-methods).

Reviewers' comments:

Reviewer's Responses to Questions

**Comments to the Author**

1. Is the manuscript technically sound, and do the data support the conclusions?

Reviewer #1: Yes

Reviewer #2: Yes

2. Has the statistical analysis been performed appropriately and rigorously? 

Reviewer #1: N/A

Reviewer #2: N/A

3. Have the authors made all data underlying the findings in their manuscript fully available?

Reviewer #1: Yes

Reviewer #2: Yes

4. Is the manuscript presented in an intelligible fashion and written in standard English?

Reviewer #1: Yes

Reviewer #2: Yes

5. Review Comments to the Author

Reviewer #1: The findings of this study present a significant argument that, within the context of China, the model and key factors proposed by the sustained commitment theory do not align with the country's socio-cultural framework. However, the author does not provide a clear discussion explaining the reasons for this inconsistency. It is recommended that a more explicit analysis be included to clarify this issue.

Furthermore, the discussion lacks comparative analysis with studies from other regions, both those with similar and differing socio-cultural contexts. Incorporating such comparisons would provide valuable insights into how the findings align with or diverge from existing research. Therefore, it is suggested that this aspect be addressed to strengthen the discussion.

Additionally, the author is encouraged to review the language for consistency and accuracy. For instance, the term "prosomal growth" (line 408) should be checked for correctness, and the term "frontline" is inconsistently formatted throughout the text—appearing as both "frontline" and "front line." It is recommended that a consistent format be used throughout the manuscript.

Reviewer #2: Assessment of the Article for Publication

General Overview

The manuscript, “Stay Committed on the Frontlines”: Sustainability of the Activism of Social Workers in Guiyang, China, is a well-structured qualitative research paper that explores the sustainability of social work activism in China, particularly in Guiyang. It provides valuable insights into the experiences of frontline social workers and challenges existing theories on sustained commitment.

The study's main contribution lies in its argument that "creativity" is essential for sustaining activism, yet it also introduces "confidence" as a prerequisite, expanding the theoretical framework of sustained commitment. The paper engages in an in-depth discussion of social workers' experiences, making an important empirical and theoretical contribution to social work research.

Strengths

1. Relevance and Contribution to Social Work and Activism Studies

The study tackles an underexplored issue: the sustainability of social work activism in China. This is especially important in a rapidly evolving social and economic landscape.

It offers an innovative extension to the sustained commitment theory by introducing "confidence" as a key prerequisite, adding depth to the field of social work activism.

The paper provides a detailed, context-specific analysis of frontline social workers, offering valuable qualitative insights.

2. Strong Theoretical Foundation

The study engages with the sustained commitment theory while also acknowledging its limitations.

It effectively critiques and expands the theory, providing a nuanced perspective on activism persistence in social work.

The use of historical and theoretical background on the evolution of social work in China strengthens the paper's argument.

3. Methodological Rigor

The qualitative approach is well-executed, with 15 in-depth interviews across diverse social work fields.

The sample represents a range of specializations (e.g., geriatric social work, environmental social work, anti-drug work), making the findings more robust.

Thematic analysis is appropriate for the research question, and the paper provides clear justification for its methodological choices.

4. Engaging and Clear Presentation

The paper is well-structured, following a logical flow from introduction to discussion.

The literature review is thorough and well-integrated into the study’s argument.

Findings are clearly presented and linked back to theoretical frameworks.

5. Ethical Considerations

Ethical approval was obtained, and informed consent was secured from all participants.

The study demonstrates a strong commitment to confidentiality and ethical research practices.

Areas for Improvement

1. Lack of Quantitative Data or Triangulation

While the qualitative approach is appropriate, the study could benefit from some quantitative elements (e.g., a survey to validate key themes found in interviews).

Triangulation with additional sources, such as policy documents or statistical reports on social worker retention in China, would strengthen the study’s claims.

2. Limited Generalizability

The research focuses only on social workers in Guiyang, which limits its applicability to other regions in China.

Given the economic and social differences across China, a comparative study between urban and rural social workers or between different provinces would improve the robustness of the findings.

3. Deeper Discussion on Policy Implications

The paper discusses social workers’ relationships with the government but does not provide enough concrete recommendations on how policies could support sustained activism.

How should policymakers and institutions address the issues of low wages, high turnover, and lack of professional recognition? More specific suggestions would make the paper more impactful.

4. Minor Language and Structural Issues

Some sentences could be clearer, and there are occasional grammatical errors (e.g., "event major life aspects such as marriage" should be "even major life aspects such as marriage").

The discussion could be better structured by explicitly separating theoretical contributions from practical implications.

Final Verdict: Suitable for Publication with Minor Revisions

The paper is a valuable contribution to social work activism studies. It is well-researched, theoretically sound, and methodologically rigorous. However, addressing the limitations mentioned above—particularly by incorporating more policy recommendations and ensuring language clarity—would enhance its impact and suitability for publication.

Recommended Next Steps

Clarify and strengthen the discussion section by explicitly stating how the findings impact theory, practice, and policy.

Consider integrating some quantitative validation (if possible in a revision or future study).

Ensure linguistic accuracy by refining sentence structure and fixing minor grammatical errors.

Expand the policy discussion to offer concrete solutions for improving the sustainability of social work activism in China.

6. PLOS authors have the option to publish the peer review history of their article (what does this mean? ). If published, this will include your full peer review and any attached files.

**Do you want your identity to be public for this peer review?** For information about this choice, including consent withdrawal, please see our Privacy Policy .

Reviewer #1: No

Reviewer #2: No

---

## [Author Response · Author response to Decision Letter 1]

23 Mar 2025

Dear reviewers and editor,

We are pleased to resubmit the revised manuscript titled "‘Stay committed on the frontlines’: sustainability of the activism of social workers in Guiyang, China" for your review. We appreciate the time and effort invested by you in evaluating my initial submission and providing valuable feedback to enhance the quality of this study.

We have carefully addressed each of the comments and suggestions, as outlined in the file "response to reviewers". We believe that these revisions have significantly strengthened this manuscript and improved its clarity, coherence, and scholarly contribution.

Thank you once again for being the reviewers and the editor of the present study.

Sincerely,

Yifu Liu

---

## [Decision Letter · Decision Letter 1]

2 May 2025

“Stay Committed on the Frontlines”: Sustainability of the Activism of Social Workers in Guiyang, China

PONE-D-25-03942R1

Dear Dr. Liu,

We’re pleased to inform you that your manuscript has been judged scientifically suitable for publication and will be formally accepted for publication once it meets all outstanding technical requirements.

Kind regards,

Tatchalerm Sudhipongpracha

Academic Editor

PLOS ONE

Additional Editor Comments (optional):

Reviewers' comments:

Reviewer's Responses to Questions

**Comments to the Author**

1. If the authors have adequately addressed your comments raised in a previous round of review and you feel that this manuscript is now acceptable for publication, you may indicate that here to bypass the “Comments to the Author” section, enter your conflict of interest statement in the “Confidential to Editor” section, and submit your "Accept" recommendation.

Reviewer #1: All comments have been addressed

2. Is the manuscript technically sound, and do the data support the conclusions?

Reviewer #1: Yes

3. Has the statistical analysis been performed appropriately and rigorously? 

Reviewer #1: N/A

4. Have the authors made all data underlying the findings in their manuscript fully available?

Reviewer #1: Yes

5. Is the manuscript presented in an intelligible fashion and written in standard English?

Reviewer #1: Yes

6. Review Comments to the Author

Reviewer #1: (No Response)

7. PLOS authors have the option to publish the peer review history of their article (what does this mean? ). If published, this will include your full peer review and any attached files.

**Do you want your identity to be public for this peer review?** For information about this choice, including consent withdrawal, please see our Privacy Policy .

Reviewer #1: No

---

## [Editor Report · Acceptance letter]

PONE-D-25-03942R1

PLOS ONE

Dear Dr. Liu,

I'm pleased to inform you that your manuscript has been deemed suitable for publication in PLOS ONE. Congratulations! Your manuscript is now being handed over to our production team.

Kind regards,

on behalf of

Dr. Tatchalerm Sudhipongpracha

Academic Editor

PLOS ONE